# Hypoxia Affects HIF-1/LDH-A Signaling Pathway by Methylation Modification and Transcriptional Regulation in Japanese Flounder (*Paralichthys olivaceus*)

**DOI:** 10.3390/biology11081233

**Published:** 2022-08-18

**Authors:** Binghua Liu, Haishen Wen, Jun Yang, Xiaohui Li, Guangling Li, Jingru Zhang, Shuxian Wu, Ian AE Butts, Feng He

**Affiliations:** 1Key Laboratory of Mariculture, Ministry of Education, Ocean University of China, Qingdao 266000, China; 2School of Fisheries, Aquaculture and Aquatic Sciences, Auburn University, Auburn, AL 36849, USA

**Keywords:** hypoxia stress, Japanese flounder, physiological response, transcriptional regulation, epigenetic modification

## Abstract

**Simple Summary:**

With global climate change and increased aquaculture production, fishes in natural waters or aquaculture systems are easily subjected to hypoxic stress. However, our understanding about their responsive mechanisms to hypoxia is still limited. Japanese flounder (*Paralichthys olivaceus*) is a widely cultivated marine economical flatfish, whose hypoxic responsive mechanisms are not fully researched. In this study, responses to hypoxia were investigated at blood physiological, biochemical, hormonal, and molecular levels. Responsive mechanisms of the HIF-1/LDH-A signaling pathway in epigenetic modification and transcriptional regulation were also researched. These results are important for enriching the theory of environmental responsive mechanisms and guiding aquaculture.

**Abstract:**

Japanese flounder (*Paralichthys olivaceus*) responsive mechanisms to hypoxia are still not fully understood. Therefore, we performed an acute hypoxic treatment (dissolved oxygen at 2.07 ± 0.08 mg/L) on Japanese flounder. It was confirmed that the hypoxic stress affected the physiological phenotype through changes in blood physiology (RBC, HGB, WBC), biochemistry (LDH, ALP, ALT, GLU, TC, TG, ALB), and hormone (cortisol) indicators. Hypoxia inducible factor-1 (HIF-1), an essential oxygen homeostasis mediator in organisms consisting of an inducible HIF-1α and a constitutive HIF-1β, and its target gene *LDH-A* were deeply studied. Results showed that *HIF-1α* and *LDH-A* genes were co-expressed and significantly affected by hypoxic stress. The dual-luciferase reporter assay confirmed that transcription factor HIF-1 transcriptionally regulated the *LDH-A* gene, and its transcription binding sequence was GGACGTGA located at −2343~−2336. The DNA methylation status of *HIF-1α* and *LDH-A* genes were detected to understand the mechanism of environmental stress on genes. It was found that hypoxia affected the *HIF-1α* gene and *LDH-A* gene methylation levels. The study uncovered HIF-1/LDH-A signaling pathway responsive mechanisms of Japanese flounder to hypoxia in epigenetic modification and transcriptional regulation. Our study is significant to further the understanding of environmental responsive mechanisms as well as providing a reference for aquaculture.

## 1. Introduction

Hypoxia is a global problem that affects the function and biodiversity of the aquatic ecosystem, where the health and well-being of aquatic organisms are often jeopardized [1,2,3]. When exposed to a hypoxic environment, fishes often show changes in macro physiology and micro molecules, such as swimming behavior, electrophysiology, hormonal profiles, biochemistry, and hypoxia-responsive factors [4,5,6,7]. More specifically, hypoxia inducible factor-1 (HIF-1) has been identified as an essential hypoxia-responsive factor in O_2_ homeostasis [8,9,10,11], which is composed of an induced subunit α (HIF-1α) and a constitutive subunit β (HIF-1β) [12]. In other words, *HIF-1β* expression does not respond to oxygen changes in cells, but *HIF-1α* expression does. Namely, HIF-1α is rapidly degraded in normoxic cells by ubiquitin–ligase complexes recognizing hydroxylated proline residues but accumulates in hypoxic cells by increasing expression and reducing degradation [9,13,14,15].

Gene expression is related to transcription factors, which can selectively activate or suppress their target gene transcription by identifying and binding to specific base sequences [16]. HIF-1 is a type of transcription factor that regulates the expression of many genes, named HIF-1 target genes, which are generally related to angiogenesis, metabolic adaptation, and survival [13,17]. HIF-1 target genes, such as genes coding for glycolytic enzymes, glucose transporters, and vascular endothelial growth factor (*VEGF*), generally contain one or more hypoxia response elements (HREs; 5-R(A/G)CGTG-3) in the promoter [9,12]. The HIF-1 could, as a transcription factor, bind to HREs to activate its target gene transcription [8]. Lactate dehydrogenase-A (LDH-A) is one of the critical enzymes catalyzing pyruvate into lactate in the process of anaerobic metabolism, and expresses mainly in skeletal muscle, liver, and lymphatic tissue [4,5,6]. Furthermore, Semenza et al., 1996 found an HRE in the mouse *LDH-A* gene promoter region and identified that the HRE was recognized by HIF-1 [18,19].

Gene expression patterns are also related to the epigenetic modification status of specific DNA sequences [20]. DNA methylation, a kind of epigenetic modification, can affect the transcriptional process. DNA hypermethylation is usually related to gene silencing, genomic imprinting, and X chromosome inactivation [20,21,22,23,24,25]. It was summarized that gene transcription was possibly suppressed by DNA methylation in two general ways: interfering directly with transcription factor binding and interfering indirectly with transcription factor binding by methylcytosine-binding protein [25]. Specifically, DNA methyltransferases add/modify methyl groups to nucleotides to change DNA structure and affect gene transcription [26,27]. CpG dinucleotides are the major targets for methyltransferase in vertebrates [20]. The methyltransferase or demethylase add or delete a methyl group in the fifth carbon atom of the cytosine residue in CpG dinucleotides to achieve methylation or demethylation [28]. Methylated modification status is very important in vertebrates [29]. Wu et al., 2018 hypothesized that DNA methylation modification may affect gene expression based on the discovery of a negative correlation between *Smyd1a* (SET and MYND domain containing 1a) expression and its DNA methylation levels [30]. Moreover, similar negative correlations were also observed in *MyoD* (myogenic determining factor), *IGF-I* (insulin-like growth factor 1), and *Follistatin* genes of Japanese flounder (*Paralichthys olivaceus*) [31,32]. The correlations between expression and methylation were also studied in other species, such as zebrafish, mouse, human, cattle-yak, and chicken [33,34,35,36,37]. In addition, findings of environmental factors (such as hypoxia, temperature, salinity) affecting DNA methylated modification levels have also been widely reported [38,39,40]. Thus, environmental factors may influence gene expression by changing DNA methylated modification status [31,41,42,43].

Skeletal muscle of fish is a vital high-quality protein source for humans [44]. Hypoxia stress affects skeletal muscle development by disturbing multiple signaling pathways [45]. Japanese flounder is an economically important marine flatfish [46]. Under the condition of high-density aquaculture, fishes easily encounter acute hypoxia stress, which affects growth, feed efficiency, and energy allocation to life activities [47]. Many researchers have worked on the physiological responses to hypoxia, but few studied hypoxia effects on gene methylated status along with transcriptional regulation, especially in fish skeletal muscle [20,48,49].

The aim of this study was to investigate hypoxia (2.07 ± 0.08 mg/L) responsive mechanisms of HIF-1 and its target gene *LDH-A* in transcriptional regulation and epigenetic modification of Japanese flounder. The results will reflect the physiological responses of Japanese flounder under this stress; reveal mechanisms of the hypoxia-related signaling pathway in methylated modification and transcriptional regulation; and explain the stress theory and guide aquaculture production.

## 2. Materials and Methods

### 2.1. Hypoxic Treatment and Sampling

Japanese flounder were collected from Qingdao HaoRuiYuan aquaculture Co., Ltd., Qingdao, China. All fishes were temporarily housed and reared in three aquaria tanks (10 ind/m^2^) with a recirculating seawater system for one week prior to the hypoxic experiment. Seawater was filtered and disinfected. Temperature of the seawater was maintained at 17.31 ± 0.02 °C (range: 17.10–17.70 °C), pH 7.49 ± 0.07 (range: 7.40–7.67), salinity 34.34 ± 0.07 ppt (range: 34.04–34.70 ppt), dissolved oxygen (DO) 7.95 ± 0.06 mg/L (range: 7.48–8.86 mg/L), and the photoperiod was 14 h light:10 h dark. All fish were fed twice (at 9:00 and 15:00) daily with KAI DO commercial sedimental compound feed (Santong, China). The contents of crude protein, crude fat, total phosphorus, and lysine in the feed were 55%, 10%, 1.5%, and 2.5%, respectively. Residual feed and feces were removed after the fish satiated. They were fasted for 24 h before hypoxic treatments. In the hypoxic treatment, DO, regulated by pumping air or nitrogen from a steel cylinder, was kept at 2.07 ± 0.08 mg/L (range: 1.84–2.36 mg/L), and was always monitored by a dissolved oxygen meter (YSI EcoSense DO200A, Yellow Springs, OH, USA); other factors (seawater temperature, pH, salinity, and density) were kept the same as those in the housing stage. It is worth mentioning that Japanese flounder grow normally when DO concentration is kept at about 6.5 mg/L, and growth rates increase and energy loss is reduced at high DO (14–16 mg/L) [47,50]. Therefore, a slightly higher concentration of 8.0 mg/L was selected as the DO concentration for the control group. The treatment group received a DO concentration of 2.0 mg/L (25% of the control), which is more than half the lethal DO concentration [51]. At least three biological repetitions (fish) and three technical repetitions (tank) were sampled after 0 (control), 1, 3, 6, 12, and 24 h acute hypoxia stress. Specifically, we anesthetized the fish with tricaine methanesulfonate (MS-222, 200 mg/L), and measured body weight (881.17 ± 20.22 g) and length (35.84 ± 0.37 cm). Blood was immediately collected from the caudal vein using disposable sterile syringes. The blood was put into two centrifuge tubes, one with anticoagulant (20 μL 15 g/L EDTA-Na_2_ solution per 1 mL blood) as anticoagulant blood, and the other without anticoagulant as non-anticoagulant blood. Subsequently, fish were immediately dissected for sampling skeletal muscle, which was promptly frozen in liquid nitrogen, and stored at −80 °C for RNA and DNA extraction. In addition, some muscle tissues were fixed in 4% paraformaldehyde to conduct double in situ hybridization (D-ISH).

### 2.2. Physiology, Biochemistry, and Hormone Measurements

#### 2.2.1. Physiological Indicators in Blood

The anticoagulant blood was used to determine the quantity of red blood cells (RBC) and white blood cells (WBC), and the concentration (g/L) of hemoglobin (HGB). Specifically, within 24 h of sampling, these indicators were detected after being shaken and mixed well by referring to Ni et al., 2016 and strictly following the instructions of BC-1800 automatic blood cell analyzer (Mindray, Shenzhen, China) [52].

#### 2.2.2. Biochemical Indicators in Serum

Non-anticoagulant blood was centrifuged (6000× *g*, 10 min, 25 °C) to obtain blood serum and then stored at −80 °C. Here, seven biochemical parameters related to metabolism were measured, which included the enzymatic activities of lactate dehydrogenase (LDH), alkaline phosphatase (ALP), alanine aminotransferase (ALT), and the concentration of glucose (GLU), total cholesterol (TC), triglyceride (TG), and albumin (ALB). According to Ni et al., 2016, we used a BS-180 automatic biochemical analyzer (Mindray, China) to collect these data and strictly followed the instructions of related kits (Mindray, China) [52].

#### 2.2.3. Hormonal Indicator in Serum

The concentration of cortisol (COR) in blood serum was detected using a GC-911 gamma radioimmunoassay counter (ZONKIA, Hefei, China) and Iodine [^125^I] cortisol radioimmunoassay kit (CNNC, Beijing, China) following the manufacturer’s instructions.

### 2.3. Genetic Structure and Phylogenetic Analysis of HIF-1α and LDH-A

The gene sequence analyses were performed with Gene Structure Display Server (GSDS 2.0) software (http://gsds.gao-lab.org/index.php, accessed on 24 April 2022). The protein physical chemical characteristics (molecular weight and theoretical isoelectric point), domains, and transmembrane helice numbers were predicted using online software ProtParam tool (https://web.expasy.org/protparam/, accessed on 24 April 2022), Simple Modular Architecture Research Tool (SMART) (http://smart.embl.de/, accessed on 24 April 2022), and TMHMM Server v. 2.0 (http://www.cbs.dtu.dk/services/TMHMM/, accessed on 24 April 2022), respectively. Preliminarily proteins’ (HIF-1α and LDH-A) three-dimensional structures were also predicted by SWISS-MODEL online software (https://swissmodel.expasy.org/interactive, accessed on 24 April 2022) and depicted using PyMOL 2.3.2 software (DeLano Scientific LLC, South San Francisco, CA, USA).

To examine the evolutionary relationships of Japanese flounder with other species, we downloaded the HIF-1α and LDH-A amino acid sequences of some species from NCBI (https://www.ncbi.nlm.nih.gov/, accessed on 24 April 2022). Phylogenetic trees were constructed using Molecular Evolutionary Genetics Analysis software (MEGA 7.0, https://www.megasoftware.net/, accessed on 24 April 2022) with Neighbor-Joining (NJ) method, in which topological stability of the trees was evaluated under 1000 bootstrap replications.

### 2.4. Double In Situ Hybridization of HIF-1α and LDH-A Genes in Skeletal Muscle

Frozen sections of muscle tissue were obtained according to Kanda et al., 2011 with minor modifications, in which skeletal muscle tissue was fixed, dehydrated, and embedded in 4% paraformaldehyde solution, 30% sucrose solution, and OCT embedding agent (SAKURA, Tokyo, Japan), respectively [53]. Afterwards, muscle tissue slices (thickness = 7 μm) were obtained using a tissue slicer (LEICA TP-1020, Wetzlar, Germany).

Digoxigenin (DIG) or Biotin (Bio) labeled RNA probes were obtained as follows. Objective DNA sequences of *HIF-1α* (535 bp) and *LDH-A* (555 bp) genes were obtained using cDNA as the template in high fidelity PCR amplification (Phanta^®^ Max Super-Fidelity DNA Polymerase Kit, Vazyme, Nanjing, China), whose products were purified by FastPure^®^ Gel DNA Extraction Mini Kit (Vazyme, Nanjing, China) and sequenced by Beijing Genomics Institute (China). Primer sequences (Appendix A) had three protective bases (cgc) added: SP6 promoter sequence (atttaggtgacactatagaagcg) in the 5′ end of the forward primer (F) and three protective bases (ccg): T7 promoter sequence (taatacgactcactatagggagaca) in the 5′ end of the reverse primer (R) after being designed by Primer Premier 5 software (Premier, Canada). Subsequently, objective DNA sequences were used as templates to transcribe the DIG-labeled *HIF-1α* RNA and Bio-labeled *LDH-A* RNA. Concretely, in the transcription system, 1 μg DNA template, 2 μL 10* RNA Polymerase Reaction Buffer (NEB, Ipswich, MA, USA), 1 μL Ribonuclease Inhibitor (TRANS, Beijing, China), 2 μL T7 RNA Polymerase (Roche, Basel, Switzerland), 2 μL 10* NTP (DIG RNA Labeling Mix 10* conc or Biotin RNA Labeling Mix 10* conc, Roche, Basel, Switzerland), and RNase free H_2_O were mixed to 20 μL.

The hybridization experiment of labeled RNA probe and target mRNA in muscle tissue slices was conducted according to the method of Kanda et al., 2011 and Li et al., 2020 [53,54]. A fluorescence microscope (ECHO RVL-100-G, San Diego, CA, USA) was used to take digital images.

### 2.5. Quantitative Real-Time PCR and Expression Analysis

TRIzol reagent (Invitrogen, Waltham, MA, USA) was used to extract total RNA from skeletal muscle. RNA concentration was measured using a nucleic acid analyzer Biodropsis BD-1000 (OSTC, Beijing, China), and its integrity was tested using agarose gel electrophoresis. Afterwards, a PrimeScript™ RT reagent kit with gDNA Eraser (Takara, Kusatsu, Japan) was used to obtain cDNA.

Quantitative real-time PCR (q-PCR) was conducted using an Applied Biosystems StepOne Plus Real-Time PCR System (Applied Biosystems, Waltham, MA, USA) with a TB Green^®^ *Premix Ex Taq*™ II (Tli RNaseH Plus) Kit (Takara, Kusatsu, Japan). Here, we determined mRNA relative expressions of *HIF-1α*, *LDH-A*, and *18S* (as a reference gene) after obtaining reasonable melting curves (Appendix A) and amplification efficiency (90–110%) [55,56]. In q-PCR, the sequences of all primers (except for *18S* primers, which referred to Huang et al., 2019) were designed using Primer Premier 5 (Premier, Canada) and specifically detected using Primer-BLAST (https://www.ncbi.nlm.nih.gov/tools/primer-blast/, accessed on 24 April 2022) as listed in Appendix A [57]. The q-PCR amplified system was 20 μL, including 10 μL TB Green Premix Ex Taq II (Tli RNaseH Plus), 0.4 μL ROX Reference Dye, 0.4 μL PCR Forward Primer, 0.4 μL PCR Reverse Primer, 2 μL cDNA (4 × diluted) template, and 6.8 μL dd H_2_O. Three biological repetitions were performed under the premise of triplicate as technical replicates. The q-PCR procedure was conducted as follows: 95 °C for 30 s, 40 cycles of 95 °C for 5 s, 60 °C for 30 s. Thereafter, the comparative threshold (2^−ΔΔCt^) method was used to calculate relative expression.

### 2.6. Transcriptional Regulation Detection of HIF-1 on the LDH-A Gene

The method of dual-luciferase reporter assay was used to verify the predicted transcriptional regulation relationship of HIF-1 on the *LDH-A* gene.

Firstly, HIF-1 transcription factors and their binding sequences on the *LDH-A* gene promoter were predicted using JASPAR online software (http://jaspar.genereg.net/, accessed on 24 April 2022).

Secondly, HIF-1α and HIF-1β expression plasmids (pc3.1~HIF-1α, pc3.1~HIF-1β) were constructed using single endonuclease digestion (BamHI (NEB, USA)) and homologous recombination. Specifically, the coding sequences (CDS) of *HIF-1α* and *HIF-1β* genes were amplified using high fidelity PCR with cDNA as template. The same sequences at both ends of the pcDNA3.1(+) plasmid digestion site were added to the 5′ ends of forward primer or reverse primer (Appendix A), respectively, in the high-fidelity PCR for homologous recombination. The pure target sequence, which was obtained using the FastPure^®^ Gel DNA Extraction Mini Kit (Vazyme, Nanjing, China), underwent homologous recombination with the linearized pcDNA3.1(+) plasmid to construct the expression plasmid. The constructed plasmid was transformed into DH5α competent cells (Vazyme, Nanjing, China) and transferred to LB (Luria–Bertani) solid medium. After the single colony was expanded in LB liquid medium and sequenced correctly (Sangon Biotech, Shanghai, China), we extracted the expression plasmid from expanded LB liquid medium using the EndoFree Mini Plasmid Kit II (TIANGEN, Beijing, China). Similarly, the reporter plasmid (pGL~LDH-A) was constructed using the above methods with the pGL3-Basic plasmid and the promoter sequence of the *LDH-A* gene. The slight difference was that the pGL3-Basic plasmid was digested using double endonuclease (SacI, HindIII (NEB, Ipswich, MA, USA)), and the promoter sequence of the *LDH-A* gene was obtained using high fidelity PCR under genomic DNA as the template.

Thirdly, plasmids were transfected into human embryonic kidney 293T (HEK293T) cells. In detail, we resuscitated (37 °C) HEK293T cells, which were cryopreserved in liquid nitrogen. The cells were cultured to 3rd or 4th generation, whose growth was normal and steady, in DMEM/High Glucose culture medium (Servicebio, Wuhan, China) with 10% fetal bovine serum (FBS) (absin, Shanghai, China) under 37 °C with 5% CO_2_. Subsequently, these cells were placed in 24-well plates (Corning, New York, NY, USA) at the concentration of 10^5^ cells per well. Plasmids were transfected into HEK293T cells with Xfect™ Transfection Reagent Kit (Takara, Kusatsu, Japan) when the confluence of cells reached 50–70%. PRL-TK plasmid (Promega, Madison, WI, USA), could express Ranilla luciferase, and was used as the control to detect transfection efficiency. Here, triplicates were conducted. The medium was changed after transfection for 4 h.

Fourthly, double-fluorescence values were detected using the SYNERGY HTX multi-mode reader (BioTek, Winooski, VT, USA) and Dual-Luciferase kit (Promega, Madison, WI, USA) after transfection for 48 h.

For identifying the specific binding sites of transcription factor HIF-1, we performed fragment deletion and directed bases mutation in the *LDH-A* gene promoter, and constructed fragment deletion reporter plasmids (pGL~Lf2, pGL~Lf1, pGL~Lf0) and base mutation reporter plasmids (pGL~Lmp, pGL~Lm2), similar to the above methods. Especially, the mutated sequences in base mutation reporter plasmids were obtained using fusion PCR, whose primer sequences are also shown in Appendix A.

### 2.7. DNA Methylation Detection and Analysis

To thoroughly understand the mechanism of acute hypoxia stress affecting biomolecules, we detected the methylated modification status of *HIF-1α* and *LDH-A* genes using bisulfite modification and sequenced methods from the perspective of epigenetics. In detail, the Marine Animals DNA Kit (TIANGEN, Beijing, China) was used to extract genomic DNA from skeletal muscle tissue. DNA concentration was measured using a nucleic acid analyzer Biodropsis BD-1000 (OSTC, Beijing, China), and its integrity was detected using agarose gel electrophoresis. The DNA was stored at −20 °C.

Bisulfite modified DNA was obtained using a BisulFlash^TM^ DNA Modification Kit (Epigentek, Farmingdale, NY, USA). Subsequently, a TaKaRa EpiTaq^TM^ HS (for bisulfite-treated DNA) Kit (Takara, Kusatsu, Japan) was used to perform methylation-specific PCR (MS-PCR) with bisulfite modified DNA as the template. In the MS-PCR, CpG islands and primers (Appendix A) were designed using the online MethPrimer design software (http://www.urogene.org/methprimer/, accessed on 24 April 2022). The MS-PCR products were separated by agarose gel electrophoresis, and then the target sequences were purified using a Midi Purification Kit (TIANGEN, Beijing, China). Purified DNA sequences were connected with pEASY-T1 vector (TransGen, Beijing, China), and transferred into *Trans*1-T1 phage resistant chemically competent cells (TransGen, Beijing, China). Three biological repetitions were performed, and about 10 clones in each individual were sequenced to calculate the methylated modification level. To evaluate the bisulfite modified efficiency and ensure the credibility of experimental results, we calculated the percentage of converted cytosines (excluding cytosines in CpG dinucleotides). The formula is as follows: the converted percentage = [number of converted cytosines (excluding cytosines in CpG dinucleotides)]/[total number of cytosines (excluding cytosines in CpG dinucleotides)] × 100.

### 2.8. Statistical Analysis

Data were expressed as mean ± standard error (M ± SE). To determine significant differences among different groups, we conducted a one-way ANOVA accompanied with the Duncan’s post hoc tests under the premises of normal distribution and variance homogeneity. *p* value was set at 0.05. It is worth mentioning that not every fish, but random individuals, were detected in the molecular experiment (q-PCR and DNA methylation detection) under less intra-group (technical repetition) differences in physiological indicators. In addition, linear regression and correlation analyses were performed to explore linear relationships between two variables (mRNA relative expression levels and methylation levels; *HIF-1α* and *LDH-A* mRNA relative expression levels). In the linear regression analysis, *R* was used to determine the strength of the correlation between two factors under the premise of a *p* value less than 0.05. The absolute value of *R* greater than 0.8, greater than 0.3 and less than 0.8, and less than 0.3 indicated that two factors were correlated strongly, weakly, and not, respectively. Here, SPSS 22.0 software (SPSS Co. Ltd., Chicago, IL, USA) and OriginPro 8.0 software (OriginLab, Northampton, MA, USA) were used for statistical analyses and graph construction.

## 3. Results

### 3.1. Effect of Acute Hypoxia on Physiology, Biochemistry, and Hormones

#### 3.1.1. Blood Physiology (RBC, HGB, and WBC)

RBC, HGB, and WBC had different fluctuating trends when hypoxia stress continued for 24 h. RBC was not significantly impacted by hypoxia treatment (Figure 1A). However, HGB first underwent a significant decline (i.e., values fell by 15.58% at 1 h and were significantly lower at 3 and 6 h) and later returned to levels like the control (*p* < 0.05, Figure 1B). WBC in the 12 h group (125.67 ± 3.78 10^9^/L) was nearly 1.6 times that of the control group (*p* < 0.05, Figure 1C).

#### 3.1.2. Blood Serum Biochemistry (LDH, ALP, ALT, GLU, TC, TG, and ALB)

##### Enzyme Activity in Blood Serum

With the prolongation of exposure to hypoxia, LDH activity increased, and reached a significantly higher level at 24 h than the control group (*p* < 0.05, Figure 2A). The ALP activity increased continuously for the first 3 h, was significantly higher in the 3 h treatment group than the control group (*p* < 0.05), and thereafter recovered (Figure 2B). However, the ALT activity did not change significantly during the 24 h experimental timeline (Figure 2C), where the maximum value occurred at 1 h (62.40 ± 28.67 U/L).

##### Concentration in Blood Serum

The concentration of GLU peaked at 1.87 ± 0.28 mmol/L in the 6 h hypoxic group (*p* < 0.05) and later recovered (Figure 3A). TC concentration followed a fluctuating trend (ranging from 6.27 ± 3.32 to 12.71 ± 0.73 mmol/L) across the experimental timeline (Figure 3B), while TG concentration significantly increased at longer hypoxia treatments (e.g., 6 and 24 h; *p* < 0.05, Figure 3C). There was no significant difference in ALB concentration among the 6 treatment groups (Figure 3D).

#### 3.1.3. COR Hormone

COR concentration increased and then decreased, where it was significantly higher in the 3 h group than the control and 24 h groups (*p* < 0.05, Figure 4).

### 3.2. Genetic Structure and Phylogenetic Analysis of HIF-1α and LDH-A

The results of gene sequence analyses are shown in Appendix A. The exon number of the *HIF-1α* gene (Gene ID: 109628061) and the *LDH-A* gene (Gene ID: 109638975) was 15 and 7, respectively. The protein sequence information is shown in Appendix A, in which five different domains (HLH, PAS, PAC, coiled coil, low complexity) were found in the HIF-1α protein and only one (low complexity) in the LDH-A protein. Furthermore, α helices, β sheets, and loop were all found in their three-dimensional structures (Figure 5).

The HIF-1α phylogenetic tree (Figure 6A) was grouped into two diverse clades (fish and other advanced vertebrates). The phylogenetic relationship of LDH-A was similar to that of HIF-1α except African clawed frog (*Xenopus laevis*) was closer to the fish clade (Figure 6B).

### 3.3. The Colocalization and Expression of HIF-1α and LDH-A

*HIF-1α* and *LDH-A* mRNA were co-located near the nucleus in muscle fiber cells by D-ISH (Figure 7).

The mRNA relative expressions of *HIF-1α* increased with extending exposure time to hypoxia and then returned to the same level as the control group at 24 h (Figure 8A). Throughout the overall hypoxia process, expression at 6 h was significantly higher than most other time groups (except 12 h) and was nearly 6.5-fold higher than the basal levels (*p* < 0.05). Moreover, there were no significant differences among the 0, 1, 3, and 24 h hypoxic treatment groups (Figure 8A). *LDH-A* mRNA relative expressions were significantly higher in the 6 h group than the control group (*p* < 0.05, Figure 8B).

Given the potential regulation of HIF-1 on the *LDH-A* gene, regression and correlation analyses were performed between their mRNA expressions. Here, we found that there was a positive correlation (*R*^2^ = 0.589, *R* = 0.767, *p* < 0.05) between *HIF-1**α* and *LDH-A* expressions (Figure 8C).

### 3.4. HIF-1 Transcriptionally Regulates the LDH-A Gene

The dual-luciferase reporter assay result (Figure 9A) showed that HIF-1 significantly upregulated the expression of the *LDH-A* gene (*p* < 0.05, 4th vs. 5th). In order to confirm HIF-1β being necessary for HIF-1, we also measured the relative luciferase activity values of the 6th group, in which expression plasmid pc3.1~HIF-1β was not added. The value of the 6th group was significantly lower than that of the 4th group (*p* < 0.05, b vs. c). However, interestingly, it was significantly higher than that of the 5th group (*p* < 0.05, b vs. a). In Figure 9B, the value in Lf2 was significantly higher than that in Lf1 (*p* < 0.05, b vs. a), which proved that the HIF-1 binding site was between −2443 and −2303. Combined with the results of JASPAR online software, we mutated the predicted binding bases (GGACGTGA, −2343~−2336). The values decreased significantly in Lmp (a vs. b) and Lm2 (a vs. b) compared with Lp and Lf2, respectively (*p* < 0.05, Figure 9C).

### 3.5. Methylation Modification and mRNA Expression of HIF-1α and LDH-A Genes

There are 18 CpG dinucleotides in the sequence of DNA methylation status measurement (Figure 10A,B) of the *HIF-1α* gene (located at −243 to −10). Some putative transcription factor binding sites (TFBSs) are located near or at these CpG dinucleotides. We marked and showed these TFBSs and corresponding transcription factors (SOX10, Sox17, Sox6, VDR, ETS1, ZNF354C, GATA2, and GATA3) in Figure 10B. The sequence of DNA methylation status measurement (Figure 10C) of the *LDH-A* gene (located at −753 to −511) contains 7 CpG dinucleotides (Figure 10D). Similar to *HIF-1α*, TFBSs and transcription factors are also noted, which include SPl1, ZNF354C, ETS1, ARNT::HIF1A (HIF-1), Arnt, Mycn, TFE3, Creb312, USF1, and USF2.

In the process of *HIF-1α* gene methylation level detection, agarose gel electrophoresis showed that the size of MS-PCR products was the same as the target amplified product (234 bp). We randomly selected 12 sequencing results of the *HIF-1α* gene to calculate bisulfite modification efficiency. Only 17 cytosines (55 cytosines outside 18 CpG dinucleotides per 234 bp) were not converted to thymine, indicating that bisulfite modification efficiency was 97.42%, which was very efficient (Appendix A shows part of the sequencing results). The overall methylation level was 33.58 ± 2.30% across all time groups (Figure 11A,B). DNA methylation level insignificantly changed when hypoxic stimulation occurred within 3 h, but then significantly declined in the 6 h group compared to that in the control and 1 h groups (*p* < 0.05, Figure 11B). Thereafter, at 12 and 24 h, DNA methylation level increased to comparable levels as the control group. In addition, methylation levels of single CpG dinucleotides first increased, decreased, and then recovered in most CpG dinucleotides (15/18) (Appendix A).

Regression/correlation analyses showed that *HIF-1α* expression was negatively correlated with its methylation level (*R*^2^ = 0.741, *R* = −0.861, *p* < 0.05) (Figure 11C). Regression/correlation analyses was also performed between *HIF-1α* expression and methylation level of single CpG dinucleotides (Appendix A). Here, eight significantly negative correlations were detected, including -218 CpG dinucleotides (*R*^2^ = 0.700, *R* = −0.837, *p* < 0.05), -188 CpG dinucleotides (*R*^2^ = 0.906, *R* = -0.952, *p* < 0.01), -145 CpG dinucleotides (*R*^2^ = 0.709, *R* = −0.842, *p* < 0.05), -137 CpG dinucleotides (*R*^2^ = 0.688, *R* = −0.829, *p* < 0.05), -135 CpG dinucleotides (*R*^2^ = 0.687, *R* = −0.829, *p* < 0.05), -116 CpG dinucleotides (*R*^2^ = 0.710, *R* = −0.843, *p* < 0.05), -78 CpG dinucleotides (*R*^2^ = 0.948, *R* = −0.974, *p* < 0.01), and -60 CpG dinucleotides (*R*^2^ = 0.753, *R* = −0.868, *p* < 0.05). Strong negative correlations at -188 and -78 CpG dinucleotides are worth mentioning (Figure 11D).

In the 243 bp sequence of the *LDH-A* gene, similar to the *HIF-1α* gene, we also randomly selected 12 sequencing results (containing 53 cytosines outside 7 CpG dinucleotides) which showed that only 11 cytosines were not converted to thymine, so that bisulfite modification efficiency was achieved to 98.27%; again, the results indicated high modification efficiency. Methylation level (9.55 ± 1.29%) was lower than the *HIF-1α* gene in all time groups (Figure 12A,B). The DNA methylation level first increased insignificantly at 1 h (*p* > 0.05), but it was significant at 3 h (*p* < 0.05). Thereafter, it significantly declined (at 6 and 12 h) and slightly increased again (at 24 h). In addition, single CpG dinucleotide methylation levels were similar to that of the *HIF-1α* gene (Appendix A).

No correlation was found between *LDH-A* mRNA expression and the 243 bp overall methylation level (*R*^2^ = 0.032). Furthermore, expression was similarly uncorrelated with 7 single CpG dinucleotides’ methylation levels (Appendix A).

## 4. Discussion

### 4.1. Response to Hypoxia Stress

HGB is a special protein that transports oxygen in RBC, which is even related to the immune system [58]. In this study, a significant decrease in HGB, but no significant increase in RBC was detected, indicating a decrease in oxygen delivery. The significant rebound of HGB and the slight retreat of RBC meant that fish were gradually adapting themselves to the hypoxic environment. WBC takes part in the process of cell-mediated immune response and phagocytosis directly in fish [59]. The fluctuating variations represented immune responses being activated in this hypoxic environment. Similarly, hypoxia also caused physiological response or adaption in Wuchang Bream (*Megalobrama amblycephala*), largemouth bass (*Micropterus salmoides*), and cavefish (*Astyanax mexicanus*) [60,61,62]. In short, hypoxia generally induced blood physiological responses in fish.

LDH, a carbohydrate metabolism enzyme, is a biomarker of stress and tissue damage [63,64]. Its enzymatic activity reflects the ability of anaerobic metabolism in organisms. Increased LDH activities in blood serum indicated that anaerobic metabolism enhanced, which was an adaptation strategy to hypoxia. The effect of changing dissolved oxygen concentrations on LDH activities were also found in cichlid (*Astronotus Ocellatus*) and rat [65,66]. All these findings and the result of *LDH-A* expression increase in skeletal muscle suggested that LDH was affected by hypoxia stress. ALP and ALT are liver marker enzymes [67]. Liver disease has also been further proved to be related to hypoxia stress in mice [68,69]. As a result, the fluctuation of ALP and ALT activities implied that these fishes suffered from liver damage under hypoxia.

Increased GLU and altered electrolyte homeostasis in blood are regarded as organism responses to environmental stress [70]. GLU was metabolized to provide energy by the glycolysis process in fish [71]. In our study, the result of GLU concentration gradually increasing when exposed to hypoxia indicates that metabolic stress responses occurred in these fishes. COR is one of the principal stress hormones in teleost fishes, and plays a significant role in osmoregulation, growth, and reproduction [72,73]. The significant increase in COR meant that our fishes were indeed affected by acute hypoxia stress. Furthermore, higher variability at 24 h was likely related to the unstable state of fishes. The same as Japanese flounder in our study, COR was also affected by hypoxia and other environmental pressure factors in rainbow trout (*Oncorhynchus mykiss*) and brown trout (*Salmo trutta*) [74,75]. Nevertheless, it has been noted that the importance of GLU is minor according to the absence of a unique mineralocorticoid in fishes, and it is the COR that has both glucocorticoid and mineralocorticoid roles [73]. This could explain why GLU concentration was not significantly changed among all time groups except for the 6 h group. Both TC and TG are related to lipid metabolism. Their fluctuations meant that lipid metabolisms were unstable. Lipid metabolites being significantly influenced by hypoxia was also found in Selenka (*Apostichopus japonicus*) and mice [2,76]. These findings implied that hypoxia stress may generally affect the homeostasis of the synthesis and degradation of lipids and their derivatives in many species.

The changes in physiological, biochemical, and hormonal indicators showed that acute hypoxic stress caused the stressful responses of Japanese flounder.

### 4.2. DNA Methylated Modification and Gene Expression

Environmental factors affect biological phenotypes by epigenetic modification in DNA methylation, DNA conformation, stability, and interactive ability with proteins [42,77]. Abnormal epigenetic modifications are related to diseases, such as cancer, cardiovascular disease, and neurodegeneration (Alzheimer’s disease, Huntington’s disease, amyotrophic lateral sclerosis) [78,79,80]. Furthermore, the epigenetic modifications could be changed by nutrition, smoking, pollution, temperature, hypoxia, and circadian rhythm [80,81,82].

Correlations between gene expression and methylation have been found in many species. For example, Goyal and Goyal (2019) found that hypomethylation was associated with upregulated expression [83]. A negative correlation between mRNA expression and DNA methylation was also discovered by Li et al., 2017 [84]. Similar negative correlations between gene expressions and methylated levels were also found in the *cyp19a1a* gene of Japanese flounder and the *SNCA* gene of humans (*Homo sapiens*) [85,86]. Overall, the above findings and our result of negative correlations between *HIF-1α* gene expression and methylation level further indicated the potential regulation of DNA methylation on gene expression. Attwood et al., 2002 concluded that DNA methylation did affect gene expression by preventing transcriptional activation [25]. However, it is puzzling that the expression of *LDH-A* was not correlated with its methylation level. We speculated that the 243 bp sequence, which we detected, was not the important regulatory related sequence. Thomson et al., 2022 found that ocular growth rates were related to single site methylation levels [87]. Therefore, combining with the strong negative correlations of -188 and -78 sites in the *HIF-1α* gene, we are speculating that the methylation status of key nucleotide sites may be more important than the overall methylation level of CpG islands.

### 4.3. Transcriptional Regulation of HIF-1 and Its Target Genes

The HREs in HIF-1 transcription factor target genes are relatively conserved like HIF-1α and LDH-A [88]. The phenomenon of HIF-1 regulating *LDH-A* gene expression was found in mice [19]. Kuo et al., 2017 discovered that HIF-1α upregulated *LDH-A* expression in human (*Homo sapiens*) hypoxic neuroblastoma cells [89]. Xie et al., 2009 also pointed out that the *LDH-A* gene was one of the HIF-1α target genes in human cells [90]. Both these reports and our results of D-ISH, i.e., the positive correlation between *HIF-1α* and *LDH-A* gene expression, all implied that HIF-1 may transcriptionally regulate *LDH-A* gene expression in Japanese flounder. However, the importance of HIF-1β in HIF-1 and the specific binding sites of HIF-1 on the *LDH-A* gene still need to be further explored in Japanese flounder.

The dual-luciferase reporter assay proved the above conjecture (Figure 9A). In addition, we also confirmed that HIF-1β was a synergistic factor for HIF-1α transcriptional regulation. The significant difference between the 5th and 6th treatments in Figure 9A was regarded as the endogenous HIF-1β interference in HEK293T cells. In brief, the results implied that the *LDH-A* gene was the HIF-1 target gene in Japanese flounder. Furthermore, the binding site of the HIF-1 transcription factor was located at −2343~−2336 (GGACGTGA) according to the results of fragment deletion and directed bases mutation experiment. Therefore, the sequence in sites of −2343~−2336 (GGACGTGA) may be the most important sequence for *LDH-A* gene sensing hypoxia. Additionally, its mutation will possibly affect transcriptional activation, and then affect a series of downstream cascade amplification effects of the signal pathway and is unfavorable to organisms or even affect the normal life activities. Here, we also conclude that not all predicted HRE sequences are the binding sites of HIF-1 transcription factors, which need to be identified by experiments rather than prediction. In conclusion, transcriptional regulation of HIF-1 on *LDH-A* may be common in organisms, but its specific nucleotide binding sites in many species need to be further studied.

Hypoxia exposure can result in large changes in gene expression, which is considered to be a response and potentially enhances hypoxic survival [91]. It has been widely reported that the target gene of HIF-1 (consisting of inducible HIF-1α and constitutive HIF-1β) includes *GLUT-1*, *Epo*, *VEGF,* and *Hsp27* [12,60,92]. The results of HIF-1α being affected by hypoxia were found in largemouth bass (*Micropterus salmoides*) [60], oscar (*Astronotus ocellatus*) [93], Atlantic croaker (*Micropogonias undulatus*) [94], and Wuchang Bream (*Megalobrama amblycephala*) [62]. The changes in HIF-1α could influence the transcription of its target genes; thus, affect the whole metabolic network. Specifically, HIF-1, as a transcription factor, can recognize and bind specific promoters of target genes, recruit RNA polymerase, turn on transcription, and then improve target gene transcription levels, resulting in a series of downstream events. Luo and Wang (2018) also concluded that HIF was related to many human diseases in physiology and pathogenesis [95]. Therefore, the important regulatory role of HIF-1 response to hypoxic stress in many species needs to be further studied in the future.

The signal networks composed of hypoxia-related signaling pathways were activated by hypoxia to adapt themselves to the hypoxic environment in cells and organisms. These pathways were mainly related to respiration, metabolism, and survival [96]. Therefore, the hypoxia signaling networks in Japanese flounder need to be further studied in the future.

## 5. Conclusions

Hypoxia caused stress responses in physiology, biochemistry, hormone, mRNA expression, and DNA methylation modification of Japanese flounder. We studied the epigenetic modification and transcriptional regulation responsive mechanisms of HIF-1α (a typical inducible hypoxia-responsive factor) and its target gene *LDH-A*. In the study, it was concluded that hypoxia affected HIF-1 by methylated modification, and HIF-1 could transcriptionally regulate its target gene *LDH-A* by binding to a hypoxia response element (HRE) of GGACGTGA located at −2343~−2336, and then LDH-A affected the downstream cascade amplification, and further influenced the physiological phenotype of Japanese flounder. These physiological and molecular changes indicated that environmental factors caused various coordinated responses at multiple levels in organisms. These findings will provide a reference for the role of the environment on biomolecules and biomolecular interactions.

## Figures and Tables

**Figure 1 biology-11-01233-f001:**
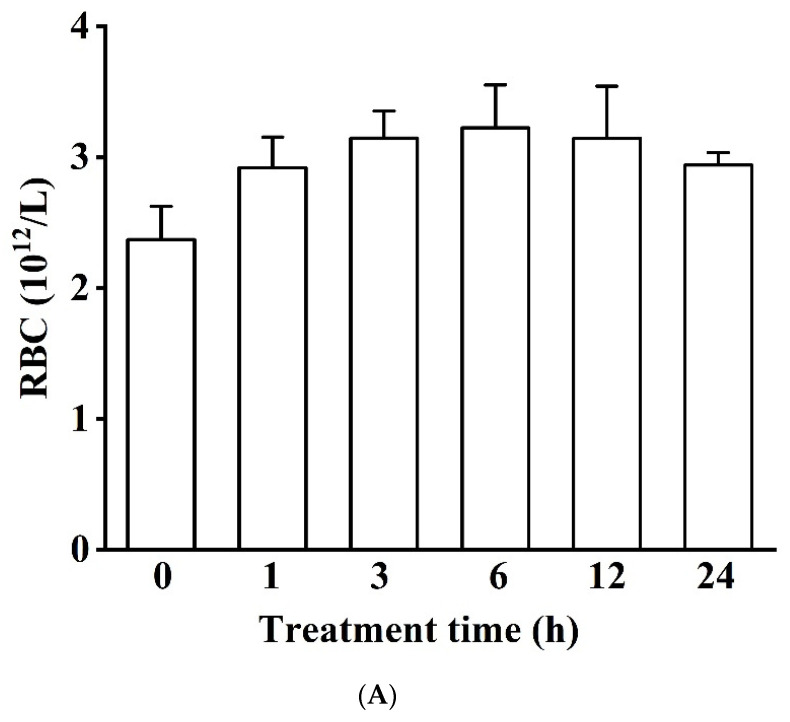
Effects of acute hypoxia on blood physiological indicators (red blood cell (RBC) concentration (**A**), hemoglobin (HGB) concentration (**B**), and white blood cell (WBC) concentration (**C**) in Japanese flounder (*Paralichthys olivaceus*). There was no significant difference in RBC concentration among the six treatment groups. Different letters indicate significant differences (*p* < 0.05).

**Figure 2 biology-11-01233-f002:**
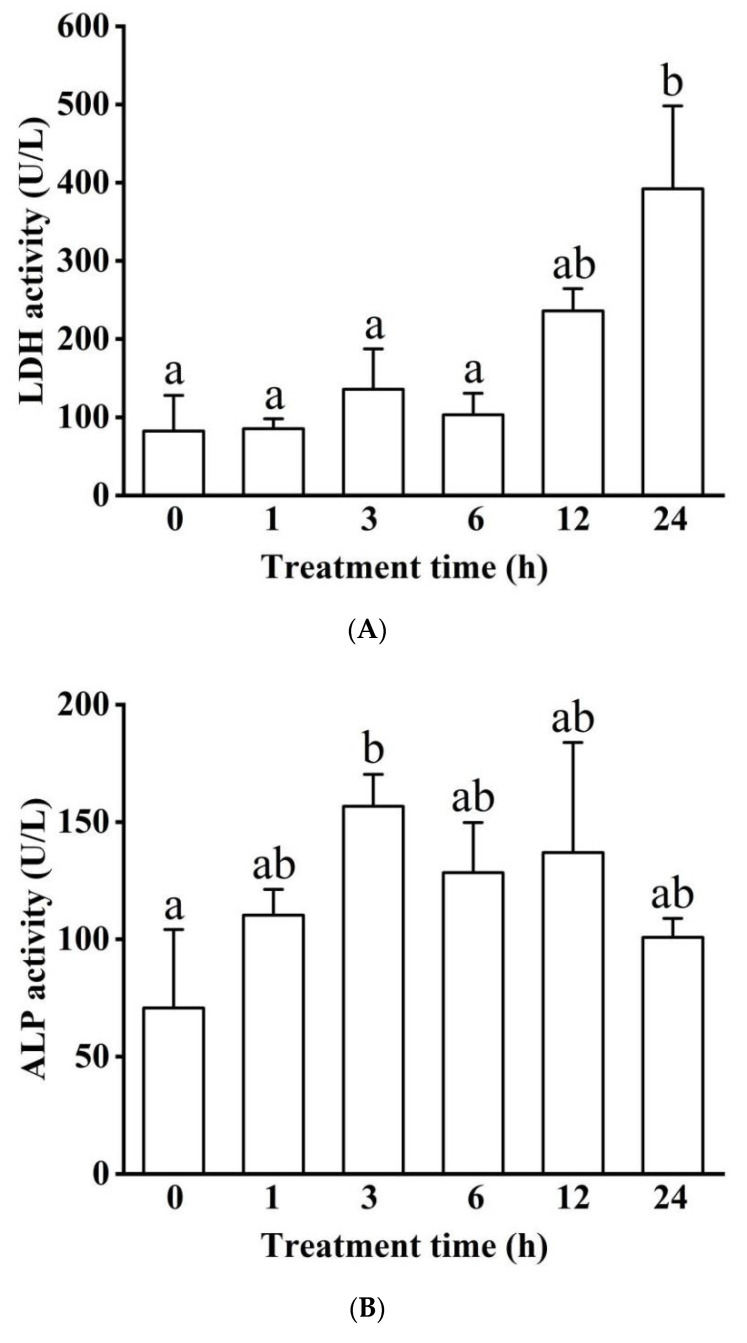
Effects of acute hypoxia on enzymatic activities of lactate dehydrogenase (LDH) (**A**), alkaline phosphatase (ALP) (**B**), and alanine aminotransferase (ALT) (**C**) in blood serum. Different letters indicate significant differences (*p* < 0.05). There was no significant difference in ALT activity among the six treatment groups.

**Figure 3 biology-11-01233-f003:**
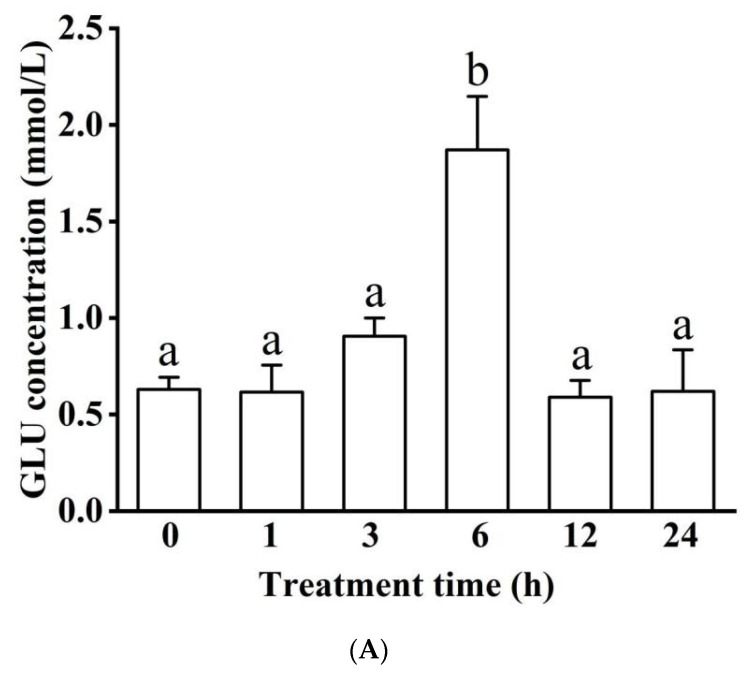
Effects of acute hypoxia on concentrations of glucose (GLU) (**A**), total cholesterol (TC) (**B**), triglyceride (TG) (**C**), and albumin (ALB) (**D**) in blood serum. Different letters indicate significant differences (*p* < 0.05). There was no significant difference in ALB concentration among the six treatment groups.

**Figure 4 biology-11-01233-f004:**
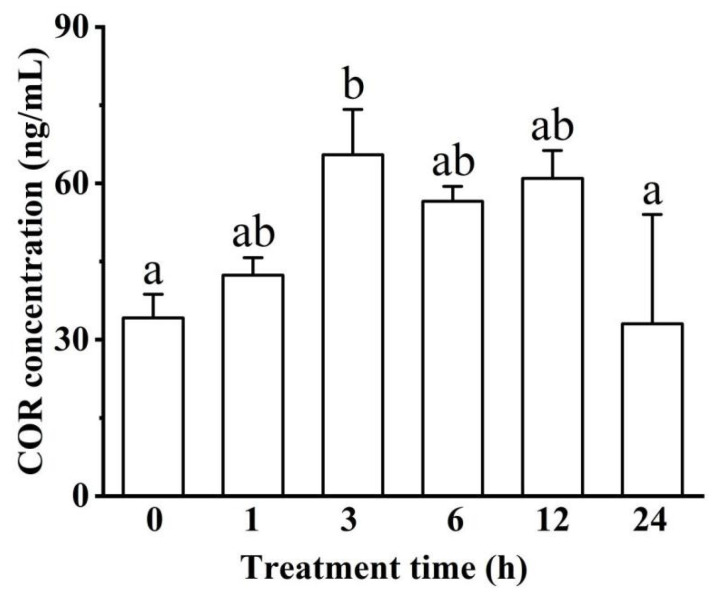
Effect of acute hypoxia on cortisol (COR) concentration in blood serum. Different letters indicate significant differences (*p* < 0.05).

**Figure 5 biology-11-01233-f005:**
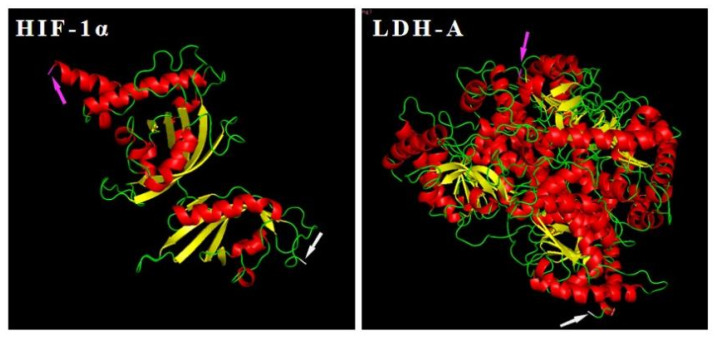
The three-dimensional structures of HIF-1α and LDH-A protein. Magenta arrows point to the N-term and gray arrows to the C-term. The α helices, β sheets, and loop are colored in red, yellow, and green, respectively.

**Figure 6 biology-11-01233-f006:**
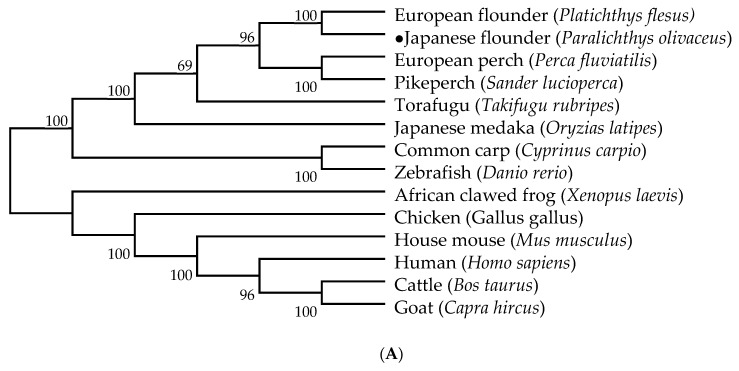
Phylogenetic trees of HIF-1α (**A**) and LDH-A (**B**). The site of Japanese flounder is marked with a solid black dot (●). The accession numbers of HIF-1α protein are: Japanese flounder (*Paralichthys olivaceus*): XP_019940496.1; Human (*Homo sapiens*): NP_001230013.1; Zebrafish (*Danio rerio*): NP_001296971.1; House mouse (*Mus musculus*): NP_001300848.1; Goat (*Capra hircus*): NP_001272657.1; African clawed frog (*Xenopus laevis*): NP_001165655.1; Chicken (*Gallus gallus*): NP_989628.1; Cattle (*Bos taurus*): NP_776764.2; European flounder (*Platichthys flesus*): ABO26720.1; Japanese medaka (*Oryzias latipes*): XP_023807064.1; European perch (*Perca fluviatilis*): ABO26717.1; Common carp (*Cyprinus carpio*): ABV59209.1; Pikeperch (*Sander lucioperca*): ABO26718.1; Torafugu (*Takifugu rubripes*): XP_003962523.2. The accession numbers of LDH-A protein are: Japanese flounder (*Paralichthys olivaceus*): XP_019957754.1; House mouse (*Mus musculus*): NP_001129541.2 Human (*Homo sapiens*): NP_001158886.1; Zebrafish (*Danio rerio*): NP_571321.1; African clawed frog (*Xenopus laevis*): NP_001081050.1; Chicken (*Gallus gallus*): NP_990615.1; Cattle (*Bos taurus*): NP_776524.1; Japanese medaka (*Oryzias latipes*): XP_023811963.1; Torafugu (*Takifugu rubripes*): XP_003967413.1; Horse (*Equus caballus*): NP_001138352.1; Tongue sole (*Cynoglossus semilaevis*): XP_024911705.1; Mexican tetra (*Astyanax mexicanus*): XP_007228114.1; Southern platyfish (*Xiphophorus maculatus*): XP_005808896.1.

**Figure 7 biology-11-01233-f007:**
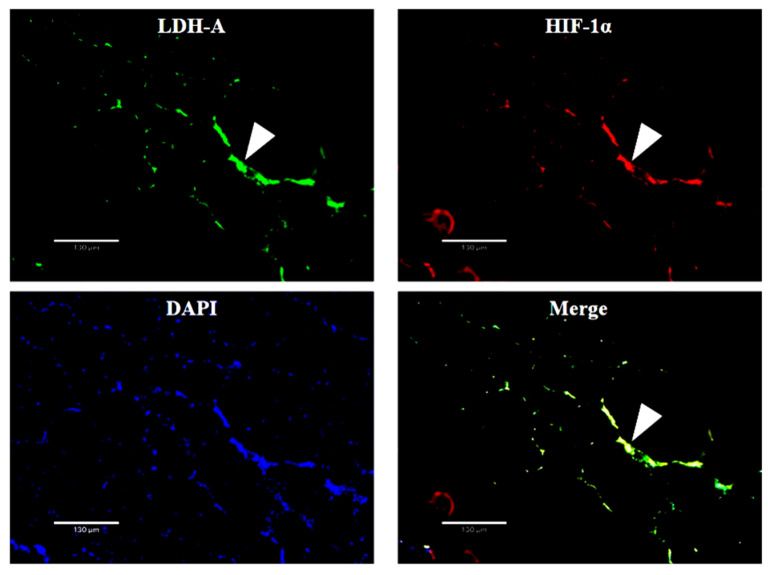
The double in situ hybridization (D-ISH) result of *LDH-A* and *HIF-1α* RNA in skeletal muscle. Green fluorescence (**upper left**) and red fluorescence (**upper right**) represent *LDH-A* mRNA and *HIF-1α* mRNA, respectively. The yellow-green fluorescence (**bottom right**) indicates the location of these two mRNA. Nuclei are shown by blue fluorescence (**bottom left**). The three arrows show that these two mRNA are located at the same position.

**Figure 8 biology-11-01233-f008:**
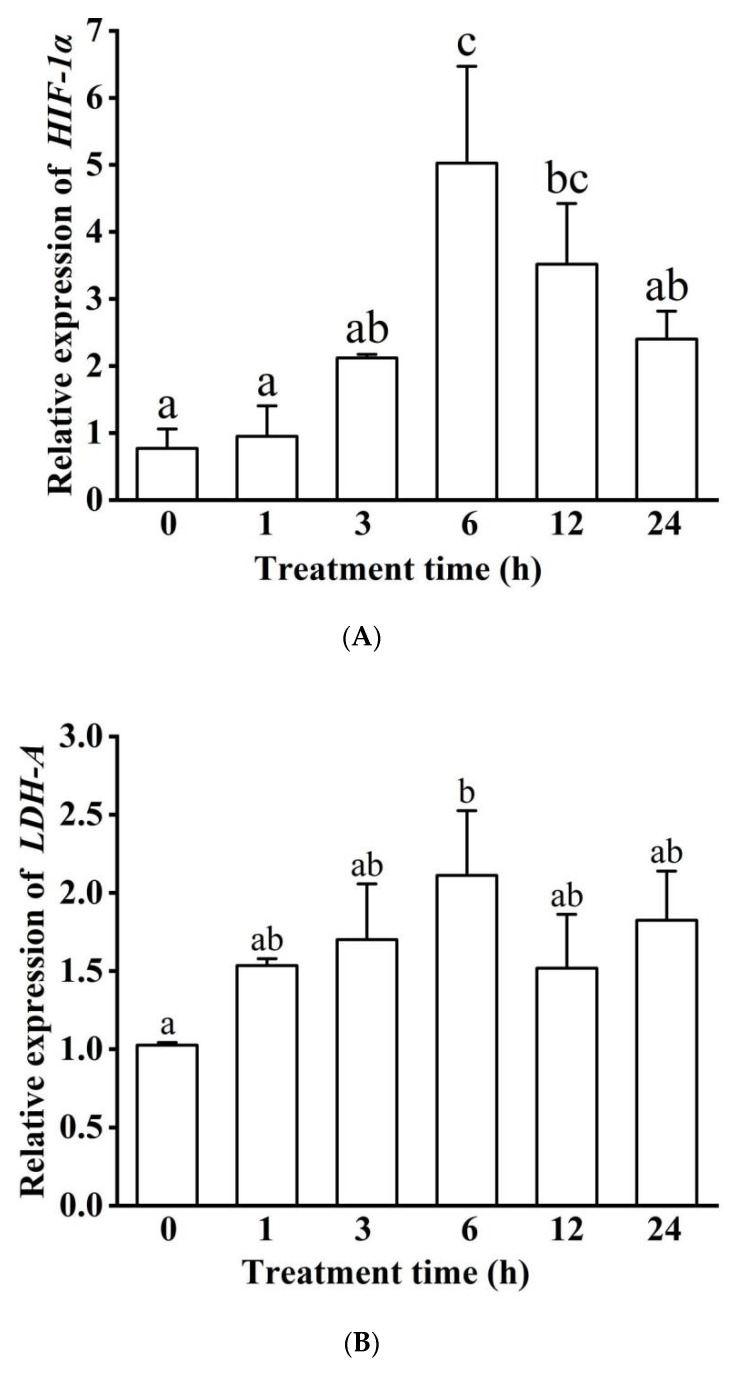
*HIF-1α* and *LDH-A* gene expression (**A**,**B**) and their correlation (**C**) under hypoxic stress. Different letters indicate significant differences (*p* < 0.05).

**Figure 9 biology-11-01233-f009:**
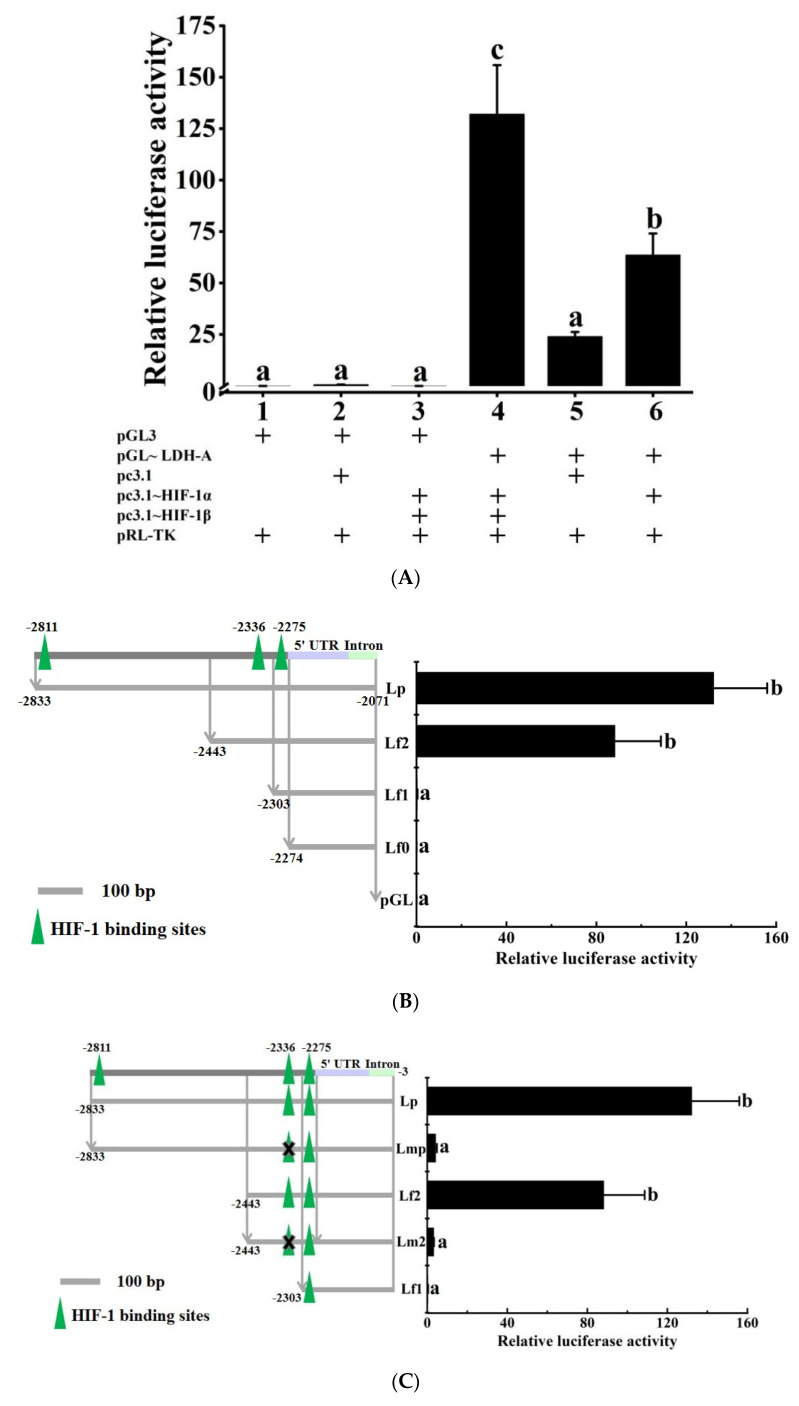
The regulatory relationship of transcription factor HIF-1 on the *LDH-A* gene in the dual-luciferase reporter assay. (**A**) Transcription factor HIF-1 transcriptionally regulates the *LDH-A* gene. The pGL3 and pc3.1 represent circular pGL3-Basic plasmid and circular pcDNA3.1 (+) plasmid, respectively, which are not linked to the exogenous sequence. Reporter plasmid pGL~LDH-A was constructed using the *LDH-A* gene promoter sequence and fragmented pGL3-Basic plasmid digested by double endonuclease (SacI and HindIII). Expression plasmid pc3.1~HIF-1α was constructed using the *HIF-1α* CDS sequence and fragmented pcDNA3.1 (+) digested by single endonuclease (BamHI). Expression plasmid pc3.1~HIF-1β was constructed using the same method. PRL-TK is the control plasmid expressing Renilla luciferase (Rluc) to characterize the transfection efficiency. Arabic numerals of abscissa in Figure 9A indicate different treatment groups. (**B**) The fragmentation deletion result. The high triangle icons represent the putative HIF-1 binding site, which is predicted using JASPAR online software. Lp, Lf2, Lf1, and Lf0 represent reporter plasmids connecting different length promoter sequences, which are shown by four light gray horizontal lines. These reporter plasmids (pGL~Lf2, pGL~Lf1, and pGL~Lf0) were connected by different length *LDH-A* gene promoter sequences and fragmented pGL 3-Basic digested by double endonuclease (SacI and HindIII), respectively. Plasmids of pGL~Lp and pGL are the same as pGL~LDH-A and pGL3 in Figure 9A, respectively. (**C**) The bases mutation result. We mutated one putative transcription factor binding site (−2336) to obtain two mutant plasmids (pGL~Lmp, pGL~Lm2), which were deleted putative HIF-1 binding sequences (−2343~−2336: GGACGTGA). The mutant plasmids were constructed using fragmented pGL3-Basic digested using double endonuclease (SacI and HindIII) and the mutant sequence obtained using high fidelity fusion PCR. Different letters indicate significant differences (*p* < 0.05).

**Figure 10 biology-11-01233-f010:**
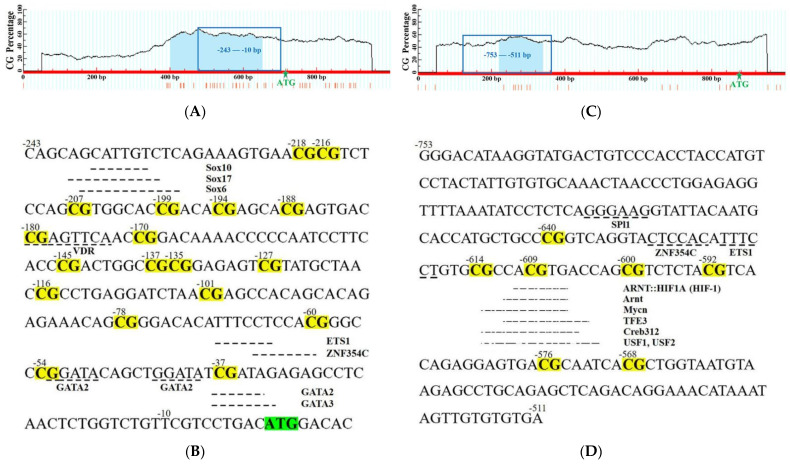
The methylation status measuring sequence analyses of *HIF-1α* and *LDH-A* genes. (**A**) The methylation status measuring region of the *HIF-1α* gene. The abscissa indicates a part of the *HIF-1α* gene; ordinate denotes CG percentage; light blue shaded part shows the CpG island (CG percentage more than 50%, −336~+219, 555 bp); blue box represents the methylation status measuring region (234 bp (from −243 to −10), with 18 CpG dinucleotides); and green five-pointed star represents the position of start codon (ATG). (**B**) The 234 bp sequence of methylation status measurement in the *HIF-1α* gene. The 18 CpG dinucleotides are indicated in yellow shading and the bold letters, and the start codon is shadowed in green and bold. The transcription factor binding sites (TFBSs) are marked with dashed underlines, and their corresponding transcription factors (SOX10, Sox17, Sox6, VDR, ETS1, ZNF354C, GATA2, and GATA3) are near the corresponding TFBSs. (**C**) The methylation status measuring region of the *LDH-A* gene. Its illustration is the same as Figure 10A except that the abscissa indicates a part of the *LDH-A* gene; the light blue shaded part shows the sequence with a CG percentage more than 40% (−653~−535, 119 bp); and the blue box represents the methylation status measuring region (243 bp (from −753 to −511), with 7 CpG dinucleotides). (**D**) The 243 bp sequence of methylation status measurement in the *LDH-A* gene. Its 7 CpG dinucleotides are indicated in yellow shading. TFBSs are shown with dashed underlines, and their corresponding transcription factors (SPl1, ZNF354C, ETS1, ARNT::HIF1A (HIF-1), Arnt, Mycn, TFE3, Creb312, USF1, and USF2) are near them.

**Figure 11 biology-11-01233-f011:**
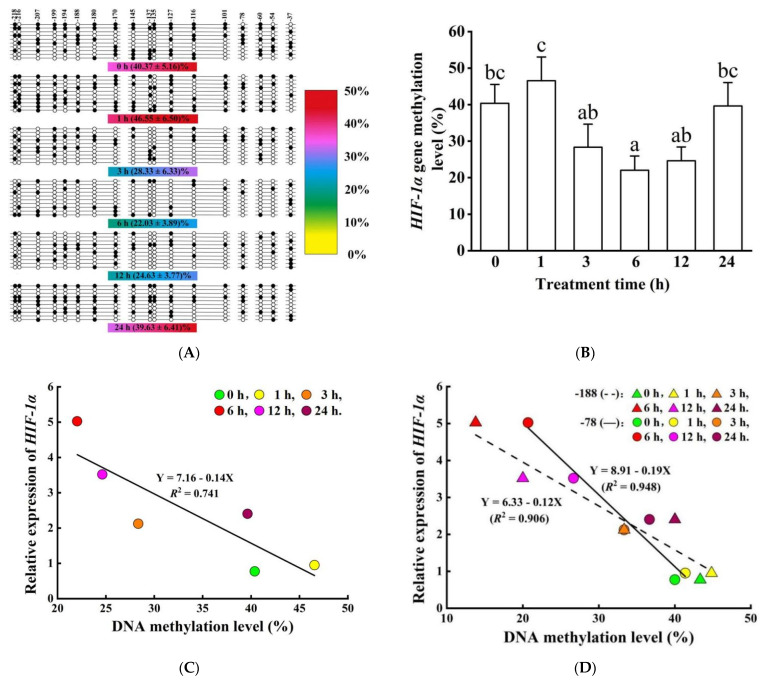
*HIF-1α* gene DNA methylation level and its correlation with expression. (**A**) The methylation status of the *HIF-1α* gene. The methylation status is represented by circles, in which filled circles show methylation and open circles represent unmethylation. The numbers above circles are the CpG dinucleotides’ locations relative to the start codon, and the percentages under the circles mean average methylation level (M ± SE) in corresponding hypoxia treatment groups. The same below (Figure 12A). (**B**) The methylation level of CpG dinucleotides in different hypoxia treatment groups. The 0 in the abscissa represents the control group. (**C**) The correlation of the *HIF-1α* gene mRNA expression and overall methylation level. (**D**) The correlations between mRNA expression level and -188 CpG dinucleotides, as well as -78 CpG dinucleotides’ methylation levels of the *HIF-1α* gene in different hypoxia groups. Different letters indicate significant differences (*p* < 0.05).

**Figure 12 biology-11-01233-f012:**
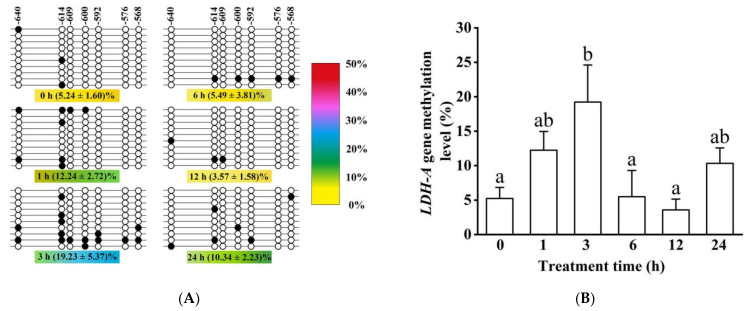
DNA methylation status of the *LDH-A* gene. (**A**) The DNA methylation status of the *LDH-A* gene. The legend is similar to Figure 11A. (**B**) The methylation level of the *LDH-A* gene in different hypoxia groups. Different letters indicate significant differences (*p* < 0.05).

## Data Availability

Data used to support the findings from this study are available from the corresponding author upon request.

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
