# Peer review of "Hypoxia Affects HIF-1/LDH-A Signaling Pathway by Methylation Modification and Transcriptional Regulation in Japanese Flounder (Paralichthys olivaceus)"

_biology, 2022, doi:10.3390/biology11081233_

Round 1

Reviewer 1 Report

Dear authors

The manuscript of Liu et al shows interesting research about hypoxia conditions in flounder. Important results were shown after analysis of several results. It is well written, nevertheless I do have some major revisions for authors to improve their manuscript.

Line 11: Please change the word inevitably, as producers try their best to prevent hypoxia in their production tanks.

Line 13: of widely cultivated

Line 41: exposed

Line 58: The HIF-1 could, as a transcription factor, bind to HREs...

Line 91: Please change the word budget, maybe metabolism?

The main revisions are in the methods and material section, especially section 2.1.

Line 111: please rewrite this, fish get satiated and were fasted for 24 hours.

Line 112-116: please rewrite this, except DO? is this the control group? the hypoxia group has a DO of 2.07. Put a full stop after the mg/L). Oxygen was monitored constantly? during the hypoxia tretament and other factors ....

Why did authors choose a DO of 2.07 as hypoxia?

Line 116-117: replicates, do authors mean by biological replicates fish or tanks? and by technical replicates, the measurements of each sample? It seems that others only have a n=3 fish. Three tanks should be used for each treatment to prevent the tank effect

Line 120-121: Please rewrite coagulant and non-coagulant blood. It is not the blood but the EDTA which is the anti-coagulant used for blood. 

Line 135: please indicate speed in xg

When presenting figures without statistical differences please do not use letters. Please remove from Fig. 1-1A, 1-2C and 1-3D

Line 427: please rewrite the sentence, was located between sequence

Line 503: correlated

Line 631: explored

line 648-650: It has been? what do authors mean?

Line 652: oscar, Atlantic croaker and Wuchang

Line 653: could influence

Reviewer 2 Report

The work submitted to me for evaluation is of a very high scientific, qualitative and innovative level. Attention is drawn to the high level of detail of the research. The work deals with the problem of hypoxia in fish, which is important from the aquaculture point of view. The minimal editorial remarks presented below do not contribute to a high evaluation of the work. The work contains all the necessary elements to be published in the journal Biology, which, after the authors respond to the comments, I hereby recommend.

Part Keywords: Suggestion to limit the number of keywords

Part Introduction: Suggestion to add information on the production volume of Japanese flounder in aquaculture. Suggestion of shortening the goal of the work and starting it with the words "The aim of this study was ..."

Materials and methods section: Suggestion to indicate the percentage of chemical composition of the feed used.

Part of the Results: I have no objections.

Part Discussion: the need to complete the year of publication after indicating the authors in all manuscript as in lines 608, 610 or 618.

Round 2

Reviewer 1 Report

Dear authors

Thank you for the revised manuscript.

I believe that the answer authors gave me to point 9: why was a DO of 2.07 choosen should be given in the manuscript. Please add this information in section 2.1 or eventually in the introduction. Perform a English grammar check to the content before adding it to the manuscript please (keeping should be kept, ....and energy loss reduce when at high DO),

Line 114. was kept at .. and always monitored by a dissolved

Line 117-119: as authors replied that they have used, correctly, three tanks per treatment this should be clarified in the manuscript. It is not clear for readers that 3 fish were sampled per tank, so authors have 9 fish per treatment for analysis. Please adapt these sentences

Line 122-124: two centrifuge tubes, one with anticoagulant (without hyphen) (20 ul...) as anticoagulant blood, and the other without anticoagulant as non-anticoagulant blood.
